# *In Silico* Discovery and Evaluation of Inhibitors of the SARS-CoV-2 Spike Protein–HSPA8 Complex Towards Developing COVID-19 Therapeutic Drugs

**DOI:** 10.3390/v16111726

**Published:** 2024-10-31

**Authors:** Liberty T. Navhaya, Thabe M. Matsebatlela, Mokgerwa Z. Monama, Xolani H. Makhoba

**Affiliations:** 1Department of Biochemistry, Microbiology, and Biotechnology, University of Limpopo, Turfloop Campus, Sovenga 7270, South Africa; 202417284@keyaka.ul.ac.za (L.T.N.); thabe.matsebatlela@ul.ac.za (T.M.M.); mokgerwa.monama@ul.ac.za (M.Z.M.); 2Department of Life and Consumer Sciences, College of Agriculture and Environmental Sciences, University of South Africa (UNISA), Florida Campus, Roodepoort 1709, South Africa

**Keywords:** COVID-19, SARS-CoV-2 spike protein, molecular chaperone, HSPA8–spike protein complex, small molecules, drug discovery, in silico

## Abstract

The SARS-CoV-2 spike protein is pivotal in the COVID-19 virus’s life cycle, facilitating viral attachment to host cells. It is believed that targeting this viral protein could be key to developing effective COVID-19 prophylactics. Using in silico techniques, this study sought to virtually screen for compounds from the literature that strongly bind and disrupt the stability of the HSPA8–spike protein complex. To evaluate the interactions between the individual proteins and the protein complex attained from protein–protein docking using BioLuminate, molecular docking was performed using the Maestro Schrodinger Suite. The screened small molecules met all bioavailability conditions, Lipinski’s and Veber’s rules, and the required medicinal chemistry properties. Protein–protein docking of the spike protein and HSPA8 identified the optimal pose with a PIPER cluster size of 65, a PIPER pose energy of −748.301 kcal/mol, and a PIPER pose score of −101.189 kcal/mol. Two small molecules, NSC36398 and NSC281245, showed promising docking scores against the spike protein individually and in a complex with HSPA8. NSC36398 had a docking score of −7.934 kcal/mol and a binding free energy of −39.52 kcal/mol with the viral spike protein and a docking score of −8.029 kcal/mol and binding free energy of −38.61 with the viral protein in complex with HSPA8, respectively. Mevastatin had a docking score of −5.099 kcal/mol and a binding free energy of −44.49 kcal/mol with the viral protein and a docking score of −5.285 kcal/mol and binding free energy of −36.65 kcal/mol with the viral protein in complex with HSPA8, respectively. These results, supported by extensive 2D interaction diagrams, suggest that NSC36398 and NSC281245 are potential drug candidates targeting SARS-CoV-2 spike protein.

## 1. Introduction

The coronavirus disease 2019 (COVID-19) pandemic severely affected the global economy and the health systems of all countries worldwide. A new beta-coronavirus called severe acute respiratory syndrome coronavirus 2 (SARS-CoV-2) [1] triggered an outbreak in Wuhan City, Hubel province, China in December 2019 [2], which led to the COVID-19 crisis. This outbreak quickly spread worldwide to other countries, being assigned as a global pandemic, the second pandemic in the 21st century [3]. According to global statistics, over 760 million people have been infected with COVID-19 since December 2019, with a mortality rate of over 6.8 million as of March 2023 [4].

The single-stranded, positive-sense SARS-CoV-2 RNA genome contains fourteen open reading frames (ORFs) that encode nine accessory proteins, sixteen non-structural proteins, and four structural proteins [5,6]. The SARS-CoV-2 spherical envelope is composed of an envelope protein (E), a spike glycoprotein (S or spike protein), and a membrane protein (M) [7,8]. Furthermore, spike proteins have been identified as potential targets for antiviral drug development. The SARS-CoV-2 spike protein is a trimeric protein with three distinct chains, chains A, B, and C, that mediate viral entry into host cells, initiating subsequent pathogenesis. The spike protein comprises two functional subunits, i.e., S1 and S2. The S1 subunit contains an N-terminal domain (NTD) and a receptor-binding domain (RBD). On the other hand, the S2 subunit consists of fusion peptides (fusion peptide 1 (FP1) and fusion peptide 2 (FP2)), a central helix (CH), heptad repeat units (heptad repeat 1 (HR1) and heptad repeat 2 (HP1)), a transmembrane domain (TM), a connector domain (DM) and a cytoplasmic tail (CT) [5,6,7,8].

The S1 subunit plays a pivotal role in viral entry and pathogenesis. It identifies and binds to host cell receptors, known as viral attachment. Subsequently, the S2 subunits facilitate the fusion of the viral-enveloped membrane and the host cell membrane, also known as membrane fusion [7,9]. Angiotensin-converting enzyme 2 (ACE2) is a well-known and well-documented host cell receptor for cellular entrance. Regardless, the virus may enter the cell via either the clathrin-independent non-endosomal pathways or clathrin-mediated endosomal pathways [10,11,12]. According to the reviewed literature, little is known about the pathogenesis of hCoVs, and what is known suggests that the level of expression of chaperones may potentially impact coronaviruses’ viral load [13]. These pro-infection and antiviral activities present opportunities for developing antiviral treatments and therapies via exploiting their immune activities or inhibiting the molecular chaperones with pro-infection activities [14].

HSP70, specifically heat shock 70 kDa 8 (HSPA8), has been reported to be involved in viral life cycle regulation, in a case known as chaperonopathy. As a constitutive heat shock protein, HSPA8 plays a vital role in protein folding, targeting proteins at the lysosome machinery for degradation and protein translocation [15,16]. The COVID-19 virus activates the host organism’s chaperonin system and commandeers the host organism’s chaperoning mechanism, altering its usual activities to favour infection. Under these conditions, HSPA8 potentially facilitates the viral attachment and endocytosis, viral penetration, uncoating, viral assembly and budding, and viral transcription and replication of SARS-CoV-2 within the host [17,18].

Recent studies postulated that human HSPA8 is localized in the cell surface membrane and melanosomes, where it serves as a membrane-anchored protein expressed in specific cell linings’ cytoplasmic membranes [17,18,19]. Human HSPA8’s location on the cell surface membrane allows the molecular chaperon to function as a receptor that binds with the SARS-CoV-2 spike proteins, mediating viral entry into host cells [17,18,19]. Additionally, HSPA8 is known to be present in both the cytoplasm and nucleus and to shuttle between these two compartments to perform various functions [20].

According to the literature, HSPA8 plays a role in the life cycle of various RNA and DNA viruses by promoting viral entry, intracellular trafficking, and disassembly and replication [13,14,15,16,17,18,19]. HSPA8 is known to be a component of the host–cell membrane receptors in the dengue virus (DENV), rotaviruses, Japanese encephalitis virus (JEV), and infectious bronchitis virus (IBV), promoting viral entry and infectivity replication [13,14,15,16,17,18,19]. HSPA8 is involved in intracellular trafficking and disassembly in JEV and has also been proven to be involved in the viral replication of the SARS-CoV-2 virus and the Ebola virus [18,21]. In the context of DNA viruses, HSPA8 has been linked to the internalization of adenoviruses (ADV), herpes simplex viruses (HSV), and hepatitis B viruses into the host cells. It is associated with the viral replication of Epstein–Barr viruses (EBV), HSV, and HBV. Moreover, HSPA8 is involved in the hepatitis C virus (HCV), where it regulates viral genome translation, virion assembly, and release, and in the human immunodeficiency virus type 1 (HIV-1) replication cycle [13,18,21].

Many studies have demonstrated that HSPA8 is a primary target that is exploited by both DNA and RNA viruses during different stages of infection replication [13,21]. Our previous molecular docking work [20], between the human HSPA8 and spike protein, showed that amino acid residues from HSPA8 (residues 403–545) and spike protein (residues 27–45) interacted with each chain, forming extensive electrostatic interactions, hydrogen bonds and hydrophobic interactions. The binding energies acquired from the docking process between chains A, B, and C from spike protein and HSPA8 were −828.3 kcal/mol, −827.9 kcal/mol, and −827.9 kcal/mol, respectively, suggesting that the interaction was spontaneous and that the complex formed was relatively stable (Appendix A) [20].

Given the pivotal role played by HSPA8 in mediating various stages of the viral life cycle, we must investigate the role played by the chaperone in the SARS-CoV-2 life cycle. Since the first step in the virus’ life cycle involves binding the spike protein to a host cell receptor, the development of an effective prophylactic against COVID-19 should preferentially target the viral spike protein [22]. However, less research has been conducted to identify potential inhibitors that have inhibitory properties against the HSPA8–SARS-CoV-2 spike protein complex (HSPA8–spike protein); hence we must investigate this gap in the literature.

One of the most plausible strategies for combating this fast-spreading pandemic was to develop vaccines against the emerging and re-emerging COVID-19 and its variants. This led to the development of over 64 vaccine candidates against SARS-CoV-2. These vaccines were developed using a variety of technologies including protein subunit vaccines, mRNA vaccines, viral vector (replication defective) vaccines, virus-like particles, and inactivated pathogen vaccines [23]. Two mRNA-based vaccines (one developed by Moderna (Cambridge, MA, USA) and the other by Pfizer-BioNTech (New York, NY, USA)) and a non-replicating viral-based vaccine (developed by Johnson & Johnson) were granted emergency use authorization. These vaccines appeared successful in containing the virus; however, due to the emergence of new SARS-CoV-2 variants of interest and variants of concern with multiple mutations within the spike protein, these vaccines’ efficacy was compromised [23].

A promising approach to address the vast array of diseases, including COVID-19, is the repurposing of clinically validated therapeutics. Most drug-repurposing studies involve synthetic chemicals; however, naturally occurring compounds can also present significant prospects [24,25]. Compared to combinatorial chemistry, natural chemicals offer remarkable structural diversity and are a crucial source of effective therapeutic leads. Natural compounds are suitable prospects for innovative therapeutics due to their diverse chemical space structures, affordability, lack of severe adverse effects, and innate biological properties [24]. Repurposing natural compounds could be a practical approach to managing SARS-CoV-2 infections.

In developing a potent COVID-19 prophylactic, targeting the viral spike protein that binds HSPA8 during the virus’s lifecycle is preferable. In our computer-aided drug design (CADD) approach, we docked four naturally occurring compounds with the potential to disrupt the HSPA8–spike protein interface interactions. Polyporic acid (NSC44175), aristolochic acid (NSC11926), 2-(3,4-Dihydroxyphenyl)-3,6,7-trihydroxy-2,3-dihydro-4H-chromen-4-one (NSC36398), and mevastatin (NSC281245) (Table 1) were previously investigated to determine whether they have inhibitory properties against the SARS-CoV-2 main protease (SARS-CoV-2 M^pro^) [26].

In the study conducted by [26], these four compounds were among the best eight selected compounds based on the highest docking scores from CDOCKER and AutoDock Vina, as well as intermolecular hydrogen bonding with the SARS-CoV-2 M^pro^ active site. They (NSC36398, NSC281245, NSC4417 and NSC11926) showed the highest binding affinity and stability in the proteases’ active site throughout the 150 ns MD simulation that was carried out, indicating their potential as leads for COVID-19 [26]. Building on these promising findings, we selected these four naturally occurring compounds for further investigation, hypothesizing that their previously demonstrated inhibitory properties against the SARS-CoV-2 M^pro^ could be repurposed to target different host–viral protein interactions, specifically the HSPA8–spike protein complex. Given their demonstrated strong binding affinity and stability towards the SARS-CoV-2 M^pro^ [26], these naturally occurring compounds are promising candidates for targeting this novel complex, thereby expanding their antiviral potential.

## 2. Materials and Methods

### 2.1. Homology Modelling and Validation of HSPA8 and SARS-CoV-2 Spike Protein Structures

In the absence of a viable high-resolution crystal structure of the SARS-CoV-2 spike protein in complex with HSPA8, we used homology modelling to build the two proteins independently. We initially built the missing regions of the spike protein 3D structure using the PDB ID:6VXX structure as a template via the SWISS-MODEL server [27]. However, the human HSPA8 crystal structure was unavailable on PDB (www.rcsb.org). Therefore, we used the SWISS-MODEL server again to generate a human HSPA8 3D structure model using Alpha fold AF-P11142-F1 as a template [28]. An unstructured section of the HSPA8 3D structure (residue positions 614–646) was removed as it was deemed unfavourable by the result from SWISS-MODEL. The quality of the structures was validated using PDBsum (https://www.ebi.ac.uk/thornton-srv/databases/pdbsum/Generate.html, accessed on 29 December 2023) [28] and Qmean (https://swissmodel.expasy.org/qmean/, accessed on 29 December 2023) [27].

### 2.2. Identification and Selection of Potential Small Molecules

Four compounds, NSC44175, NSC11926, NSC36398, and NSC281245 (Table 1), were selected from a previous study [26]. The four small molecules are known to have inhibitory properties on the SARS-CoV-2 M^pro^. All molecules were obtained directly from PubChem, except for mevastatin, which was obtained from DrugBank (https://go.drugbank.com/drugs/DB06693, accessed on 21 November 2023) [29]. For the screening of the small molecules, websites and software like PubChem (https://pubchem.ncbi.nlm.nih.gov/, accessed on 21 November 2023) [30], ChEMBL (https://www.ebi.ac.uk/chembl/, accessed on 21 November 2023) [31], and SWISS ADME (https://www.swissadme.ch, accessed on 21 November 2023) [24] were used to verify the drug-likeness, water solubility, physiochemical properties, and medicinal chemistry of the small molecules. For the molecular docking, the small molecules were obtained directly from PubChem in ‘.SDF’ format and converted to ‘.mol2’ format using PyMOL software (version -2.5.5).

### 2.3. Molecular Docking

#### 2.3.1. Performing Protein–Protein Docking.

The interactions between SARS-CoV-2 spike protein and HSPA8 were predicted using Bioluminate 4.6 (Maestro13.1, Schrodinger suite) [32,33]. Before protein processing and docking, chain A of HSPA8 was renamed “chain H” to differentiate it from the spike protein chain A and band C. Both proteins were prepared using the “Protein Preparation Workflow” of Bioluminate 2.4, with missing loops filled using “Prime” minimized at the forcefield of OPSL4. Upon docking using “Protein–Protein Docking” under Biologics, the HSPA8 was regarded as the receptor, whilst the spike protein was regarded as the ligand. The number of ligand rotations to be probed was set to 70,000, and the maximum number of poses per return was set to 30. Following the calculation, the resulting poses were manually analyzed. The interactions between HSPSA8 and the spike protein from the best pose were analyzed using the “Protein Interaction Analysis” under Biologics [33].

#### 2.3.2. Protein–Ligand Docking

Protein–ligand docking was performed using Maestro13.1 from Schrödinger to assess the interaction between the individual proteins (human HSPA8 and the SARS-CoV-2 spike protein) and the four small molecules [34,35]. Before protein–ligand docking, the 3D protein structures of both proteins were protonated using H++ (http://newbiophysics.cs.vt.edu/H++/, accessed on 21 November 2023) [36] at a physiological pH of 7.4. Docking-ready structures were generated using the “Protein Preparation Workflow” of the Schrodinger Suite, and the resulting structures were energy-minimized using an OPLS4 forcefield. Potential binding sites were identified using SiteMap (Schrodinger Suite) [34,35], where the top five binding sites were generated. The small molecules were prepared using “LigPrep”, on of Schrodinger’s tools. Grid boxes were generated based on the most favourable binding site identified from the previous SiteMap results. The Glide programme (Schrodinger Suite) with Extra Precision (XP) was used to dock both proteins with all four small molecules. The results were manually visualized using the 2D diagrams generated from each Glide job [35].

#### 2.3.3. Performing Protein Complex–Ligand Docking.

To assess the effects of the small molecules on the HSPA8–spike protein complex attained from the previous protein–protein docking, protein–ligand docking was performed using Maestro13.1 from Schrödinger [34,35]. Similar steps were taken as those from the first protein–ligand docking. Before protein–ligand docking, the HSP8–spike protein complex was prepared using “Protein Preparation Workflow” in settings identical to the protein–ligand docking. Potential binding sites were identified using SiteMap (Schrodinger Suite), and small molecules were prepared using “LigPrep” [34,35] as previously achieved. Grid boxes were generated based on the most favourable binding site identified from the SiteMap results. The Glide programme (Schrodinger Suite) with XP was used to dock the complex with all four small molecules. Finally, the results were manually visualized for the interactions between proteins and the small molecules using the 2D diagrams generated from each Glide job [34,35].

### 2.4. Prime MM-GBSA

The binding free energies of all the protein–ligand complexes attained from docking were conducted in Prime MMGBSA. The Prime module of the Maestro Schrodinger Suite with the OPLS4 forcefield and Prime’s VSGB2.1 solvation model were exploited to calculate the binding free energy [37,38]. The binding free energy of a ligand to a protein to form the complex is depicted as follows.
∆*G _binding_* = ∆*G* _(*complex*)_ − ∆*G* _(*Protein*)_ − ∆*G* _(*Ligand*)_
where *G _binding_* is the binding free energy, *G*
_(*complex*)_, *G _(Protein)_*, and *G _(Ligand)_* represent the free energy of the complex, protein, and ligand, respectively [37,38,39].

## 3. Results

### 3.1. Screening and Analysis of Drug-Likeness Properties of the Selected Small Molecules

All the small molecules targeted in this study followed Lipinski’s rule of five and Veber’s rule with no violations. Furthermore, the small molecules satisfied all bioavailability conditions and required medicinal chemistry properties, physiochemical properties, water solubility, and drug-likeness, as listed in Table 2 [34]. The results from an ADMET analysis (https://vnnadmet.bhsai.org/, accessed on 21 November 2023) [34,40] indicate that none of the small molecules have any cytotoxicity effects (Appendix A). This ADMET analysis revealed that all four small molecules have no blockade effects on the human ether-a-go-go-related gene (hERG) potassium channel. The maximum recommended therapeutic dose (MRTD) for polyporic acid is 196 mg/day, 209 mg/day for aristolochic acid, 2092 mg/day for 2-(3,4-dihydroxyphenyl)-3,6,7-trihydroxy-2,3-dihydro-4H-chromen-4-one, and 65 mg/day for mevastatin [41].

### 3.2. Three-Dimensional Homology Modelling and Validation

SWISS-MODEL (https://swissmodel.expasy.org/, accessed on 21 November 2023) [27] was used for the homology modelling of the 3D structures of the human HSPA8 and SARS-CoV-2 spike protein, and the models were selected among the other models based on QMEANDisCo Global score and GMQE (Global Model Quality Estimate) values. GMQE and QMEANDisCo Global give an overall model quality measurement between 0 and 1, with higher numbers indicating higher expected quality [27]. The GMQE of the selected SARS-CoV-2 spike protein model was 0.71, and the QMEANDisCo Global value was 0.73 (Figure 1a) [27]. The GMQE of the selected HSPA8 model was 0.89, and the QMEANDisCo Global value was 0.80. Therefore, both values indicated a high structure quality (Figure 1b).

The protein models were validated using PDBsum (https://www.ebi.ac.uk/thornton-srv/databases/pdbsum/Generate.html, accessed on 21 November 2023) [28], PROCHECK (https://saves.mbi.ucla.edu/, accessed on 21 November 2023) [42,43], and Qmean (https://swissmodel.expasy.org/qmean/, accessed on 21 November 2023) [27]. The PDBsum results, as shown by the Ramachandran plot in Figure 2, indicate that about 94.2% of the HSPA8 amino acid residues are in the most favourable regions. The results for the SARS-CoV-2 spike protein indicate that about 90.4% of the amino acid residues are in the most favourable regions. As per Ramachandran plot analysis, a high-quality model with excellent stereochemical characteristics in the central region of Ramachandran plots is expected to exhibit a 90% proportion of residues residing in the most favoured area. Based on that knowledge, it can be concluded that the modelled structures of HSPA8 and SARS-CoV-2 spike protein are high-quality models [27,42]. Qmean validation for HSPA8 and spike protein gave QMEANDisCo global values of 0.80 (for HSPA8; Appendix A) and 0.73 (for spike protein; Appendix A), which are considered high values, indicating that the modelled structures are of high quality [27].

### 3.3. Protein–Protein Docking

Protein–protein docking using BioLuminate4.6 [32] gave an output of the thirty best docking poses (Appendix A), and the best docking pose was selected based on PIPER cluster size, PIPER pose energy, and PIPER pose score. The best-selected protein–protein docking pose (prot-prot-docking_2_pose_1) had a PIPER cluster size of 65, a PIPER pose energy of −748.301 kcal/mol, and a PIPER pose score of −101.189 kcal/mol (Appendix A) [31]. To study the interactions of the selected docking pose of the HSPA8–spike protein complex, the Protein Interaction Analysis tool from BioLuminate Schrödinger was used [31]. Appendix A was generated, showing the interactions that occur within the HSPA8–spike protein complex. From this table (Appendix A), the specific interactions were selected and are displayed in Table 3 [31].

Examination of the HSPA8–spike protein complex revealed that 88 interactions were formed (Appendix A), with a total of 7 specific interactions being formed (Table 3). The amino acid residues involved in the protein–protein interactions from HSPA8 ranged from residue position 405 to position 495, corresponding to the substrate binding domain (SBD) identified at positions 349–509 [20]. From the selected HSPA8–spike protein complex, the trimeric protein used chains A and B in the interaction. The amino acid residues involved in the protein–protein interactions from chain A of the spike protein ranged from position 455 to position 493, and those of chain B of the spike protein ranged from position 333 to position 371 (Appendix A). Both positions from each chain correspond to the C-terminal domain (CTD) of the SARS-CoV-2 spike protein [20,28]. These positions also correspond to the RBD, integrin-binding, and receptor-binding motifs, which play crucial roles in interacting with the ACE2 receptor and, in this instance, HSPA8 [20,29].

To better comprehend and visualize the chemical interactions between the SARS-CoV-2 spike proteins and human HSPA8, a 2D diagram was generated for the proteins (Figure 3) using the Protein Interaction Analysis tool from BioLuminate Schrödinger, with the interactions occurring as depicted in Table 3.

Multiple hydrogen bonds were captured in the analysis of the interactions. These interactions ranged from 1.6 to 2.5 Å, and the overall distance range was from 1.6 to 3.9 Å (Appendix A). These distance ranges suggest close proximities between the interacting amino acid residues, which is relevant for stabilizing the HSPA8–spike protein complex interface [20,31,44].

THR411 from HSPA8 formed a strong hydrogen bond with THR478 from the spike protein’s chain A with a bond distance of 1.6 Å, HSPA8’s GLN426 hydrogen bonded with SER477 from the spike protein’s chain A (distance of 1.9 Å), and ARG469 formed a 2.0 Å salt bridge with chain A’s GLU487 (Figure 3 and Table 3) [20,44]. Additionally, HSPA8‘s ARG469 formed a hydrogen bond (distance of 2.1 Å) with GLN493 from the spike protein’s chain A, while GLN473 hydrogen-bonded with ASP364 from chain B (distance of 2.2 Å). ASP433 formed a hydrogen bond with a distance of 2.4 Å with GLN493 from chain A. LYS493 from HSPA8 formed a hydrogen bond with THR333 from chain B of the spike protein with a bond distance of 2.5 Å (Figure 3 and Table 3). These interactions are pervasive; therefore, more energy will be required to break these bonds, making the protein complex formed from the interaction more stable [31,44].

### 3.4. Protein–Ligand Docking

Protein–ligand docking was performed for both HSPA8 and the spike protein to assess the protein–ligand interactions. This provided a more refined understanding of the interaction dynamics, and enhanced understanding of their potential efficacy and specificity. The binding sites were considered based on the following criteria: sitescore > 1, volume > 225, and dscore > 1 [34]. The key feature generated by SiteMap is an overall SiteScore, which has proven useful and effective in identifying binding sites [34]. SiteScore is based on a weighted sum of several properties, including the number of site points, hydrophilicity in charged and highly polar sites, and enclosure. Other properties also considered included the size of the site, the degrees of exposure and enclosure by the protein, and the hydrophilic and hydrophobic character of the sites (Appendix A) [33]. Based on the SiteMap results (Appendix A), Figure 4 shows the best potential docking sites chosen (for both proteins) for protein–ligand docking as they obeyed the specified criteria. These binding sites were then used in targeted protein–ligand docking; unlike blind docking, this approach provides computational efficiency and demonstrates the capacity to yield reliable estimates for identifying active sites [34,45]. The amino acid residues that make up the binding sites for these proteins are summarized in the Appendix A.

LigPrep was used to prepare the small molecules before docking. The conformations with the lowest Epik state penalties were chosen for those that produced different conformations, i.e., NSC36398 gave three outputs. NSC44175 had a state penalty of 0.0011 kcal/mol, NSC11926 had a state penalty of 0 kcal/mol, NSC36398 had a state penalty of 0.0325 kcal/mol, and NSC281245 had a state penalty of 0 kcal/mol (Figure 5) [46]. Epik state penalties estimate the free energy that is required in the generation of ionization states in water. For state penalties, the lower the state penalty, the better the ligand conformations. Given the above values, it can be concluded that the conformations selected from LigPrep are quite stable and viable for protein–ligand docking.

The prepared ligands were subjected to flexible docking using the XP docking programme [35] and the results indicating the docking scores, compound ID (CID), and Glide Gscores from the docking simulations are summarized in Table 4 for both proteins.

The small molecules bound to HSPA8 via interactions with the various amino acid residues in the binding site, including ASP10, THR13 TYR25, LYS71 GLU175, GLY202, LYS271, ARG272, GLY339, SER340, ARG342, and ASP366. Of the four small molecules, NSC36398 exhibited the best docking score of −7.148 kcal/mol and a Glide Gscore of −7148 kcal/mol (Table 4) [47]. Notably, NSC36398 formed four hydrogen bonds with the amino acid residues TYR25, GLU175, ARG272, and THR204 (Figure 6a), contributing to its strong binding affinity [48]. NSC36398 showed a favourable binding free energy of −37.73 kcal/mol from the binding free energy calculations, indicating strong binding free energy, which in turn suggests strong binding and stable interactions with HSPA8. The small molecule NSC281245 had the second lowest docking score of −5.244 kcal/mol and a Glide Gscore of −5.244 kcal/mol compared to NSC36398. However, the binding free energy for NSC281245 is −13.08 kcal/mol, suggesting a lower binding affinity and interaction stability than NSC36398. NSC281245 formed notable interactions, including hydrogen bonds with the amino acid residues ASP10, THR14 and LYS71. These interactions possibly account for its favourable docking score with HSPA8 compared to NSC11926 and NSC44175 (Figure 6b).

NSC11926 exhibited a docking score and a Glide Gscore of −4.141 kcal/mol (Table 4). Its binding free energy of −31.64 suggests a more substantial binding free energy relative to NSC281245. This indicates that NSC11926 potentially has a better binding affinity towards HSPA8. Further analysis of the docking of NSC11296 indicated that this small molecule forms two hydrogen bonds with the amino acid residues SER340 and GLY202, three salt bridges with the amino acids LYS271 and ARG272, and lastly, a single pi-cation bond with the amino acid residue ARG342 (Figure 6c) [38,49]. Lastly, NSC44175 showed relatively weak binding affinity compared to the other small molecules, with a docking score of −2.406 kcal/mol and a Glide Gscore of −2.407 kcal/mol (Table 4). Its binding free energy of 14.2 kcal/mol indicates an unstable interaction with the host HSPA8 protein, though it forms two hydrogen bonds and two salt bridges with the amino acid residues LYS271 and ARG272 (Figure 6d and Appendix A).

Docking simulations were also performed to assess the potential efficacy and specificity of the small molecules with the SARS-CoV-2 spike protein. The docking results revealed intriguing insights into the binding interactions (Table 4; Figure 7). The spike protein formed interactions with the small molecules using key amino acid residues, as follows: A: GLU1017; A: ASN1039; B: SER1021; B: THR1023; B: ARG1039; C: ARG1019; C: ASN1023; and C: ARG1039. Of the four molecules, NSC36398 exhibited the best binding affinity with a docking score of −7.934 kcal/mol and a Glide Gscore of −7.965 kcal/mol, plus a binding free energy of −39.52 kcal/mol, which indicates a favourable and stable interaction (Table 4). NSC36398 formed two key hydrogen bonds with the amino acid residues GLU1017 from chain A and THR1027 from chain B, which contributed to the binding affinity and further confirmed the stability of the interaction (Figure 7a). It is imperative to note that NSC36398 demonstrated a greater binding affinity and interaction stability with the viral spike protein, which outperforms its interaction with human HSPA8, with a docking score of −7.148 kcal/mol and a binding free energy of −37.73 kcal/mol.

The small molecule NSC281245 demonstrated the second-best docking score of −5.099 kcal/mol and a Glide Gscore of −5.099 kcal/mol (Table 4). The binding free energy of −44.49 kcal/mol for NSC241285 signified a high free energy release, indicating a stable interaction. Analysis of the 2D structure of the interaction indicated that the NSC281245 forms two key hydrogen bonds that potentially contribute to its binding affinity and interaction stability with ASN1023 from chain C and ARG1019 (Figure 7b). NSC281335 exhibits significantly better binding stability with the spike protein, making it a potential drug candidate.

The small molecule NSC11926 exhibited a docking score of −3.463 kcal/mol and a Glide Gscore of −3.463 kcal/mol with the spike protein (Table 4). Compared to its interaction with HSPA8 (docking score of −4.141 kcal/mol and binding free energy of −31.3 kcal/mol) human protein), NSC11926 exhibits less favourable interactions and binding free energy with the spike protein. No visible interactions were determined from the 2D diagrams, which correspond to the lower binding stability of the small molecule (Figure 7c). The absence of any specific interactions in the 2D diagram suggests a possibility in which NSC11926 engages in hydrophobic interactions rather than specific hydrogen bonds or salt bridges [47,50]. The small molecule NSC44175 showed the highest docking score of −2.873 kcal/mol, a Glide Gscore of −2.873 kcal/mol, and the least favourable binding free energy of −7.23 kcal/mol (Table 4). This indicates a low binding affinity towards the spike protein and a poor binding stability to the spike protein.

NSC44175 forms a single hydrogen bond with ARG1039 from chain C and two salt bridges with ARG1039 from chain A and ARG1039 from chain C (Figure 7d). NSC44175 also forms pi-cation interactions with ARG1039 from chains A and B. The docking score and Glide Gscore are lower than those depicted in the docking interactions with the HSPA8 protein. However, both values in each instance are lower when compared to other small molecules, indicating a lower binding affinity.

### 3.5. Protein Complex–Ligand Docking

The HSPA8–spike protein complex was subjected to SiteMap analysis to identify the binding sites within the protein complex. The top five binding sites from the protein complex were all located on the SARS-CoV-2 spike protein (Figure 8), with site 4 being the best binding site because it had the highest site score, the highest dscore, and the highest volume (Appendix A). The depicted potential binding site was then used in performing targeted protein complex–ligand docking [33,34]. The amino acid residues that comprise binding site 4 of the protein complex are summarized in the Appendix A. It can, however, be noted that the additional amino acid residues THR723, PHE759, LEU767, LEU952, LEU953, GLN1005, and VAL1008 contributed to the formation of the binding site of the complex compared to the same binding site selected for the individual spike protein. Also, VAL781 and LEU959 from the individual spike protein were not present in the binding site for the HSPA8–spike protein complex.

The prepared small molecules were similarly subjected to flexible docking using the XP docking programme [35]. The docking scores and Glide Gscores from the docking simulations are summarized in Table 4 for the complex with the four small molecules. Figure 9 shows the 2D diagrams generated to better assess the protein complex–ligand interactions.

The docking results for the HSPA8–spike protein complex demonstrated varying binding affinities amongst the four naturally occurring compounds. The small molecules docked within the binding site of the HSPA8–spike protein complex and interacted differently with the following amino acid residues: A: ARG1014; A: THR1027; A: ASN1023; A: ARG1039; B: ARG1039; and C: ARG1039. NSC36398 had the most negative docking score of −8.029 kcal/mol, a Glide Gscore of −8.061 kcal/mol, and a binding free energy of −38.61 kcal/mol (Table 4), indicating the highest binding affinity and a stable interaction. The performance of NSC36398 is consistent with its favourable binding to the human protein HSPA8 alone (docking score of −7.148 kcal/mol and binding free energy of −39.52 kcal/mol) [37,38]. NSC36398 exhibits a better binding affinity and interaction stability in relation to the HSPA8–spike complex than its binding to the individual spike protein. It forms two hydrogen bonds with amino acid residues THR1027 from chain A and GLU1017 from chain C, contributing to its binding affinity and interaction stability (Figure 9a).

NSC281245 had the second most negative docking score of −5.285 kcal/mol) and a Glide Gscore of −5.285 kcal/mol, followed by the small molecule NSC11926, which had the third most negative docking score of −4.12 kcal/mol and a Glide Gscore of −4.12 kcal/mol (Table 4). NSC44175 had the least negative docking score of −2.798 kcal/mol and Glide Gscore of −2.798 kcal/mol, indicating the weakest binding affinity towards the HSPA8–spike protein complex. NSC281245 exhibited a favourable docking score, indicating a better binding affinity towards the protein complex compared to NSC11926 and NSC44175. An analysis of the binding free energy showed that NSC281245 exhibited a binding free energy of −36.65 kcal/mol, indicating better binding stability compared to those of NSC11926 with a binding free energy of −27.16 and NSC44175 with a binding free energy of 1.61 kcal/mol, which indicates relatively weak or non-existing binding stability (Table 4). Compared to its performance with the individual human HSPA8 protein (docking score of −5.285 kcal/mol, binding free energy of −13.08 kcal/mol) and the spike protein alone (docking score of −5.099 kcal/mol, binding free energy of −44.49 kcal/mol), NSC281245 indicated improved binding stability with the protein complex (Table 4).

Two-dimensional diagrams were generated and analyzed (Figure 9). Figure 9b shows the 2D diagram of the HSPA8–spike protein complex in interaction with NSC281245. NSC281245 formed a single hydrogen bond with ALA1016 from chain A of the spike protein in complex with HSPA8. This single hydrogen bond contributed to the binding affinity and stability of NSC281245. NSC11926 formed a single hydrogen bond with the amino acid residue ARG1039 from chain C and two salt bridges, one with the amino acid residue ARG1039 from chain B and one with an amino acid residue from chain C (Figure 9c and Appendix A). Salt bridges are weak bonds compared to hydrogen bonds (Figure 9c). Hence, NSC11926 had a low binding affinity compared to NSC36398 and NSC281245. Despite NSC11926 having three bonds, its docking score and binding free energy suggest that it has a weak binding affinity and stability towards the protein complex. NSC44175 exhibited a single salt bridge interaction with ARG1014 from chain A (Figure 9d), corresponding to this small molecule’s low binding affinity and stability.

## 4. Discussion

The current study used in silico approaches to screen for small molecules that strongly bind and interact with the previously investigated HSPA8–spike protein complex, potentially exerting inhibitory effects towards the protein complex. The primary objective of the SARS-CoV-2 virus upon infecting a host is to replicate and generate numerous copies of itself exclusively [42]. Reports indicate that the virus cannot initiate this replication process independently, hence hijacking the host’s biomolecules. From the literature, the SARS-CoV-2 spike protein is known to interact with human HSPA8, where it is used for viral entry, protein refolding, and protein translocation [20]. According to recent studies, viral loads are influenced by the expression levels of molecular chaperones. Therefore, this study sought to screen and analyze small molecules that strongly bind and interact with the HSPA8–spike protein complex to exert inhibitory properties towards the protein complex.

HSPA8 is a constitutive heat shock protein that plays a role in protein folding, translocation, and targeting proteins to lysozyme machinery for degradation [20]. HSPA8 is postulated to mediate viral attachment and endocytosis, viral penetration and uncoating, viral assembly and budding, and viral replication and transcription in a virus’s life cycle [13,17,18]. These functions are made possible because of the localization of the HSPA8 in the cell surface membrane, melanosomes, cytoplasm, and the nucleus. Within the cell surface membrane and melanosomes, HSPA8 acts as a membrane-anchored protein expressed in the cytoplasmic membranes of distinctive cell linings that include smooth muscle cells, respiratory tract cells, and the epithelia of the small intestine [13,20].

The HSPA8 consists of two functional domains: an NBD and an SBD. The NBD functions as an energy supply via ATP hydrolysis, and the SBD binds to the substrate and precedes the protein folding and translocation process. As a case in point, the chaperon’s SBD (from residue positions 405 to 495) interacts with the SARS-CoV-2 spike protein’s CTD that consists of the RBD and the RBM (Appendix A, Figure 1 and Appendix A) [20,42]. The folding process is complex and requires strong binding between the heat shock protein of interest and the substrate protein. This study established that a strong complex exists between the HSPA8–spike protein due to the short bonds formed with bond lengths ranging from 1.6 to 3.9 Å (Appendix A) and the extensive specific interactions between the two proteins.

The RBD of the SARS-CoV-2 spike protein is responsible for viral attachment to the host’s cell receptors during viral entry. The RBD of the spike protein is part of the S1 subunit that performs a vital role in the viral attachment during the initial stages of viral infection [20]. The spike protein established contacts within the S1 subunit of the spike protein to form multiple interactions with HSPA8. The regions observed to form contacts correspond to the receptor-binding domain (RBD), integrin-binding motif, and receptor-binding motif, which play crucial roles in interacting with the receptor angiotensin-converting enzyme 2 (ACE2) and, in this instance, HSPA8 [20,34].

To understand whether the formation of the HSPA8–spike protein complex can potentially be disrupted, we first determined the ability of the small molecules to favour interaction with the spike protein and HSPA8. The proteins were docked with existing inhibitors attained from an article by Romeo et al., (2021) [24]. These compounds were previously investigated to determine their inhibitory properties towards the SARS-CoV-2 M^pro^. From our study, all four compounds were also docked with an already formed HSPA8–spike protein complex to determine the interactions between the protein complex and the small molecules. From the docking results, the small molecules mevastatin (NSC281245) and 2-(3,4-dihydroxyphenyl)-3,6,7-trihydroxy-2,3-dihydro-4H-chromen-4-one (NSC36398) had favourable interactions, targeting the SARS-CoV-2 spike protein.

Interestingly, the small molecule NSC36398 had docking scores of −7.148 kcal/mol with human HSPA8, −7.934 kcal/mol with SARS-CoV-2 spike protein and −8.029 kcal/mol with HSPA8–spike protein complex (Table 4) [51]. It also exhibited a binding free energy of −37.73 kcal/mol with human HSPA8, −39.52 kcal/mol with the viral spike, and −38.61 kcal/mol with the HSPA8–spike protein complex. From the results, the small molecule NSC36398 formed favourable interactions across all targets, with the highest binding affinity observed for the HSPA8–spike protein complex. This indicates that NSC36398 can interact strongly with the HSPA8–spike protein complex, suggesting its potential to exert an inhibitory effect on the complex’s formation and stability [37,38]. The binding free energy further supports the docking results, showing favourable binding free energies across all targets [52]. The more negative binding free energy for the spike protein suggests a favourable and stable interaction with the spike protein alone compared to HSPA8 and the HSPA8–spike protein complex. Still, the overall affinity is consistently high in all three targets [37,38].

Additionally, the small molecule NSC281245 had docking scores of −5.224 kcal/mol with HSPA8, −5.099 kcal/mol with the SARS-CoV-2 spike protein, and −5.285 kcal/mol with the HSPA8–spike protein complex [46] and had a binding free energy of −13.08 kcal/mol with human HSPA8, −44.49 kcal/mol with the viral spike protein, and −36.65 kcal/mol with the HSPA8–spike protein complex [37,38]. It exhibits lower docking scores than NSC36398, indicating a weaker binding affinity across all three targets. The docking scores are relatively similar, suggesting no significant preference. However, the binding free energy revealed a mixed interaction profile. The interaction of NSC281245 with the viral spike protein exhibited a more negative binding free energy score, suggesting a higher affinity towards the viral protein [38]. The interaction with the HSPA8–spike protein complex is relatively strong, but that with HSPA8 alone is the weakest of them all. This suggests that NSC281245 may potentially be more effective as a direct inhibitor [38]. Overall, the results indicate that these two small molecules effectively target the viral protein as an individual protein and in complex with the human HSPA8, making them potential drug candidates with a promising specificity, therapeutic index, and broad applicability.

## 5. Conclusions

Repurposing of small molecule inhibitors explored in previous in silico, in vitro, or other studies offers a promising approach to addressing a vast array of diseases, including COVID-19. The current study utilized in silico techniques to screen for small molecules (used previously to target the SARS-CoV-2 M^pro^) that strongly bind and interact with the previously investigated HSPA8–spike protein complex and exert potential inhibitory properties towards the protein complex. Using sophisticated software such as BioLuminate4.6 (Schrodinger Suite), it was concluded that the SARS-CoV-2 spike protein forms extensive interactions with the human heat shock protein HSPA8, which is known as a housekeeper, and is known to mediate viral attachment and endocytosis, viral penetration and viral uncoating, viral assembly and budding, and viral replication and transcription. Molecular docking simulations with four small molecules revealed that mevastatin (NSC281245) and 2-(3,4-dihydroxyphenyl)-3,6,7-trihydroxy-2,3-dihydro-4H-chromen-4-one (NSC36398) had higher docking scores with the spike protein than with human HSPA8, indicating a stronger binding affinity towards the viral protein. Further molecular docking studies on the HSPA8–spike protein complex revealed that these two small molecules target the viral protein within the complex, showing high docking scores and indicating a high binding affinity. These results show that NSC281245 and NSC36398 appear to be potential drug candidates that target the spike protein individually and in complex with human HSPA8, suggesting promising specificity (for the small molecule NSC281245), a good therapeutic index, and broad applicability (for the small molecule NSC36398). The Prime MM-GBSA energy results further supported the findings obtained from molecular docking simulations, as both NSC36398 and NSC281245 showed more favourable binding free energies. Future research perspectives focus on validating these results by first performing molecular dynamic simulations to gain a detailed and dynamic view of the interactions at the atomic level, followed by essential in vitro experimentation to confirm these findings.

## Figures and Tables

**Figure 1 viruses-16-01726-f001:**
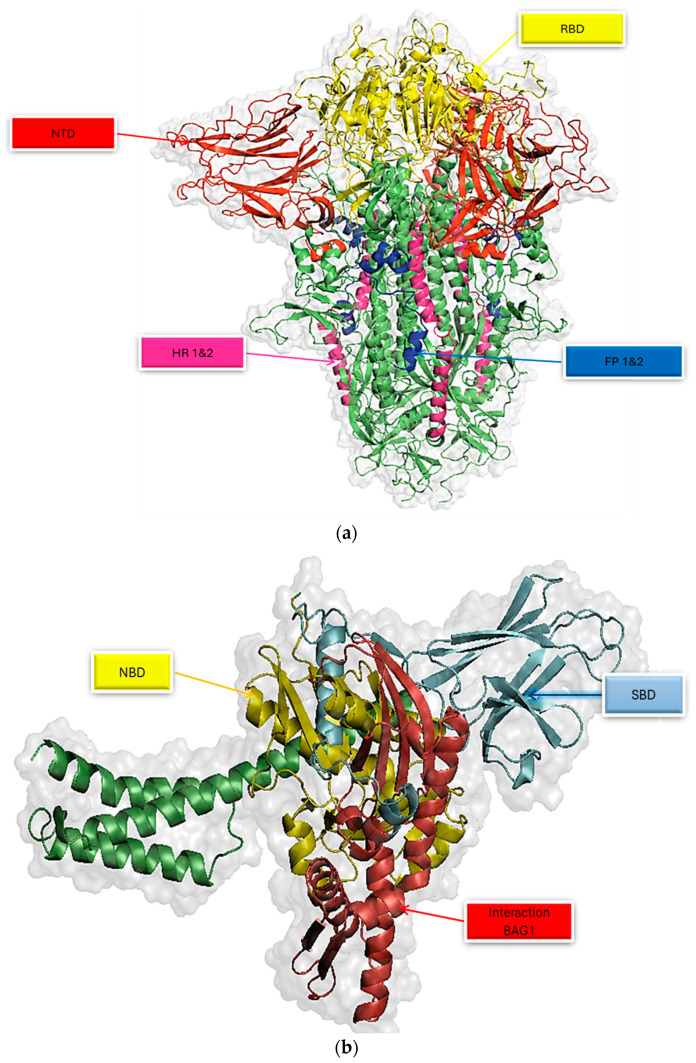
(**a**) Three-dimensional structures of the SARS-CoV-2 spike protein and human HSPA8 a. Three-dimensional structure of the SARS-CoV-2 spike protein attained from the SWISS-MODEL and visualized by PyMOL indicating the receptor-binding domain (RBD; residue positions 319–541), C-terminal domain (residue positions 334–527), N-terminal domain (NTD; residues 14–303), fusion peptides (FP1 and FP2), and heptad repeat units (HR1 and HR2). (**b**) Three-dimensional structure of human HSPA8 obtained from the SWISS-MODEL as visualized by PyMOL indicating the N-terminal nucleotide-binding domain (NBD; residue positions 2–386) and substrate-binding domain (SBD; residue positions 349–509).

**Figure 2 viruses-16-01726-f002:**
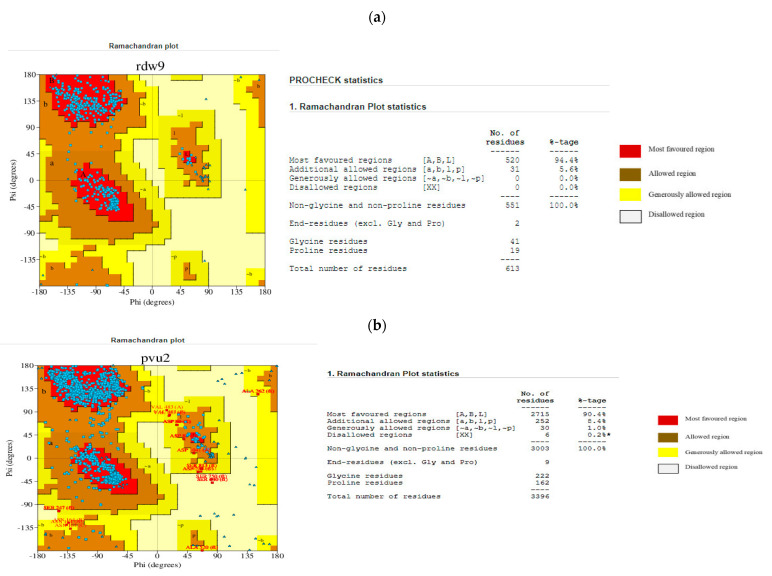
(**a**) Ramachandran plots of the modelled human HSPA8 obtained from PDBsum [28,42]. (**b**) Ramachandran plots of the modelled SARS-CoV-2 spike protein obtained from PDBsum. The asterisk (*) indicates residues in disallowed regions [28,42].

**Figure 3 viruses-16-01726-f003:**
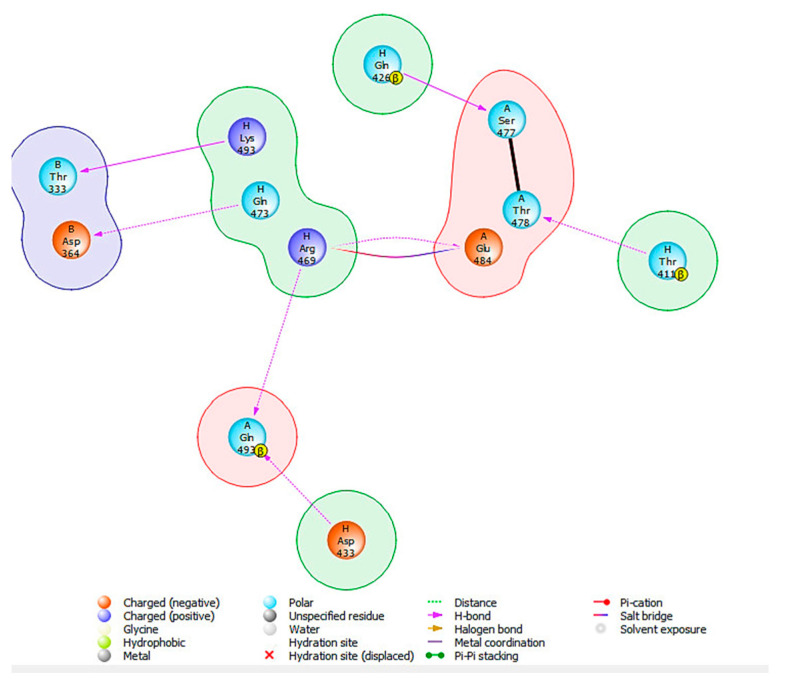
Two-dimensional diagram showing the HSPA8–spike protein complex interactions and the amino acid residues involved in forming the complex’s interface. β indicates the beta-carbon (βC) of amino acids involved in interactions with another protein.

**Figure 4 viruses-16-01726-f004:**
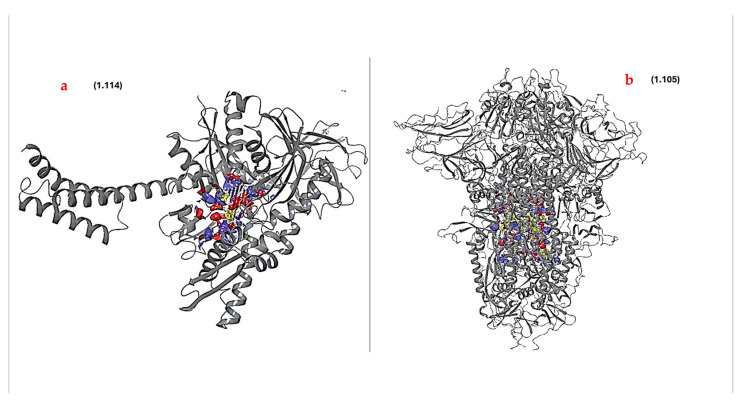
Identified best binding sites ((blue-red-yellow sections) of (**a**) human HSPA8 and (**b**) SARS-CoV-2 spike protein identified using SiteMap. The values in brackets are the site scores for the identified binding sites [35,45].

**Figure 5 viruses-16-01726-f005:**
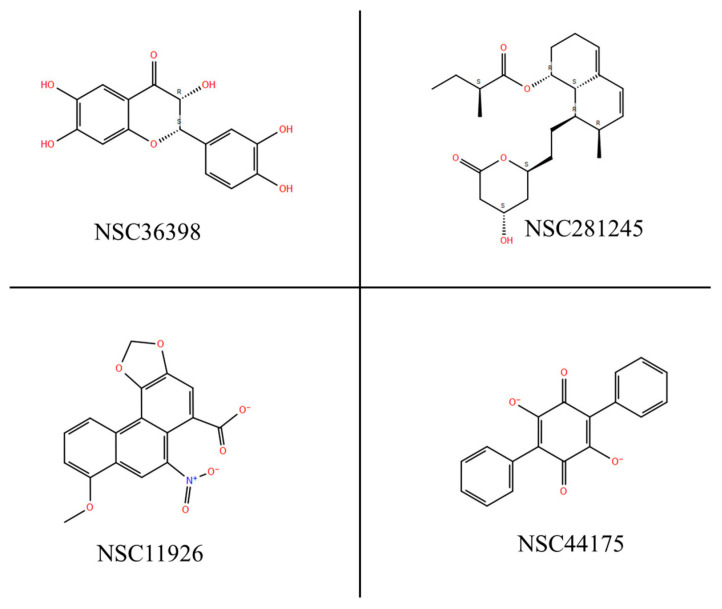
Structures of the small molecules were prepared using LigPrep [34,35].

**Figure 6 viruses-16-01726-f006:**
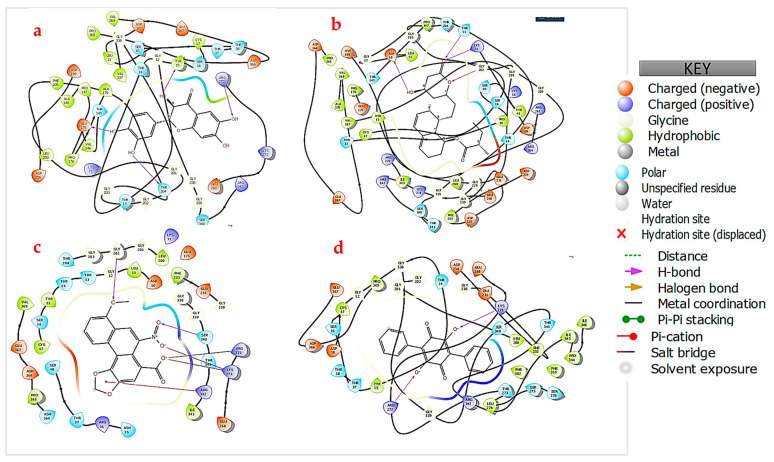
Ligand interaction diagrams. (**a**) HSPA8–NSC44175; (**b**) HSPA8–NSC11926; (**c**) HSPA8–NSC36398; (**d**) HSPA8–NSC281245.

**Figure 7 viruses-16-01726-f007:**
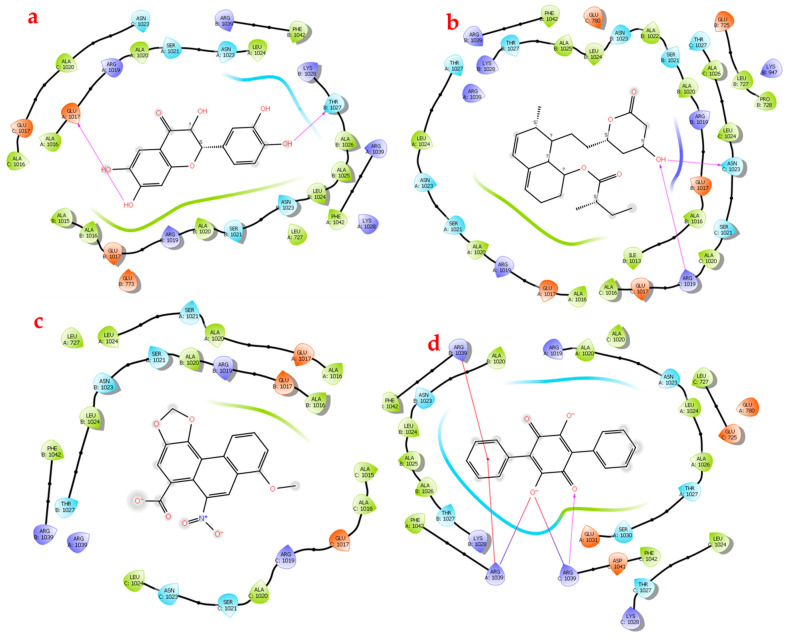
Ligand interaction diagrams. (**a**) SARS-CoV-2 spike protein–NSC44175; (**b**) SARS-CoV-2 spike protein–NSC11926; (**c**) SARS-CoV-2 spike protein–NSC36398; (**d**) SARS-CoV-2 spike protein–NSC281245.

**Figure 8 viruses-16-01726-f008:**
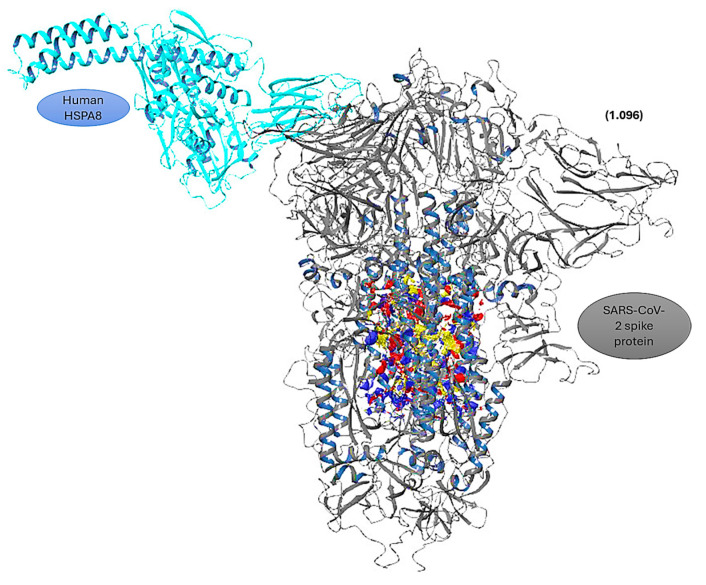
The best-identified binding site (blue-red-yellow section) of the HSPA8–spike protein complex identified using SiteMap. The value in brackets is the SiteScore [34,35].

**Figure 9 viruses-16-01726-f009:**
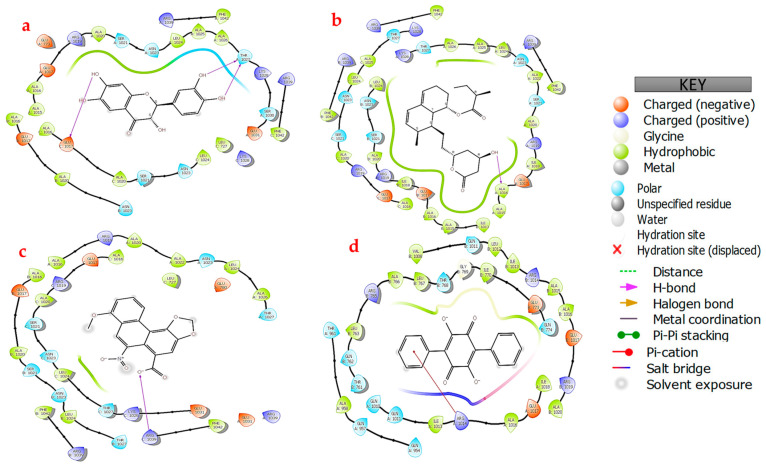
Ligand interaction diagrams. (**a**) Protein complex–NSC36398; (**b**) protein complex- NSC281245; (**c**) protein complex–NSC11926; (**d**) protein complex–NSC44175.

**Table 1 viruses-16-01726-t001:** The targeted small molecules’ structures: NSC44175, NSC11926, NSC36398 and NSC281245.

Compound Name	Structure
**Polyporic acid** NSC44175	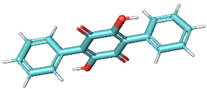
**Aristolochic acid** NSC11926	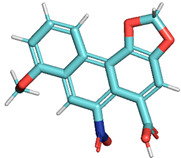
**2-(3,4-Dihydroxyphenyl)-3,6,7-trihydroxy-2,3-dihydro-4H-chromen-4-one** NSC36398	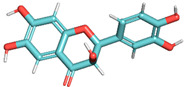
**Mevastatin** NSC281245	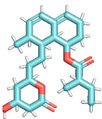

**Table 2 viruses-16-01726-t002:** Physiochemical, drug-likeness, water solubility properties, medicinal chemistry, and toxicity.

Name of Ligand	Polyporic Acid	Aristolochic Acid	2-(3,4-Dihydroxyphenyl)-3,6,7-trihydroxy-2,3-dihydro-4H-chromen-4-one	Mevastatin
**Physiochemical properties**				
Molecular formula	C_18_H_12_O_4_	C_17_H_11_NO_7_	C_15_H_12_O_7_	C_23_H_34_O_5_
Hydrogen Bond Donor Count	2	1	5	1
Hydrogen Bond Acceptor Count	4	7	7	5
Topological Polar Surface Area	74.60 Å^2^	110.81 Å^2^	127.45 Å^2^	72.83 Å^2^
Fraction CSP^3^	0.00	0.12	0.13	0.74
**Water Solubility**				
Log *S* (SILICOS-IT)	−5.14	−4.325	−2.03	−3.04
Class	Moderately soluble	Moderately soluble	Soluble	Soluble
Solubility	2.11 × 10^−3^ mg/mL; 7.21 × 10^−6^ mol/L	1.52 × 10^−2^ mg/mL; 4.47 × 10^−5^ mol/L	2.87 mg/mL; 9.42 × 10^−3^ mol/L	3.55 × 10^−1^ mg/mL; 9.08 × 10^−4^ mol/L
**Drug likeness**				
Lipinski Rule	Yes; 0 violations	Yes; 0 violations	Yes; 0 violations	Yes; 0 violations
Veber (GSK) Rule	Yes	Yes	Yes	Yes
Egan (phatmacial) Filter	Yes	Yes	Yes	Yes
Muegge (Bayer) Filter	Yes	Yes	Yes	Yes
Bioavailability (Abbort) Score	0.85	0.56	0.55	0.55
**Medicinal Chemistry**				
Pan Assay Interference Structures	1 alert: quinone A	0 alert	1 alert: catechol A	0 alert
Brenk	1 alert: chinone A	3 alerts: nitro group, oxygen-nitrogen single bond, polycyclic_aromatic_hydrocarbon_3	1 alert: catechol	1 alert: more than 2 esters
Lead likeness	Yes	No; 1 violation: XLOGP3 > 3.5	Yes	No; 2 violations: MW > 350, XLOGP3 > 3.5
Synthetic accessibility	3.00	2.77	3.52	5.56

**Table 3 viruses-16-01726-t003:** Specific interactions between the SARS-CoV-2 spike protein and human HSPA8 [25,31].

HSPA8 Residues	SARS-CoV-2 Spike Protein Residues	Distance (Å)	Specific Interactions	No. of Hydrogen Bonds
H: THR411	A: THR478	1.6	1× hb to A: THR478	1
H: GLN426	A: SER477	1.9	1× hb to A: SER477	1
H: ARG469	A: GLU487	2.0	1× hb, 1× salt bridge to A: GLU484	1
H: ARG469	A: GLN493	2.1	1× hb to A: GLN493	1
H: GLN473	B: ASP364	2.2	1× hb to B: ASP364	1
H:ASP433	A: GLN493	2.4	1× hb to A: GLN493	1
H: LYS493	B: THR333	2.5	1× hb to B: THR333	1

hb means hydrogen bonds.

**Table 4 viruses-16-01726-t004:** Docking scores, Glide Gscores, and Prime MM-GBSA binding energies of the selected compounds with HSPA8, SARS-CoV-2 spike protein, and the HSPA8–spike protein complex [35,37].

Protein	CID	Docking Scores (kcal/mol)	Glide Gscore (kcal/mol)	Prime MM–GBSA Complex Energy (*dG _bind_*) (kcal/mol)
HSPA8	NSC36398	−7.148	−7.148	−37.73
HSPA8	NSC281245	−5.224	−5.224	−13.08
HSPA8	NSC11926	−4.141	−4.141	−31.30
HSPA8	NSC44175	−2.406	−2.407	14.20
Spike protein	NSC36398	−7.934	−7.965	−39.52
Spike protein	NSC281245	−5.099	−5.099	−44.49
Spike protein	NSC11926	−3.463	−3.463	−23.90
Spike protein	NSC44175	−2.873	−2.873	−7.23
HSPA8–spike protein	NSC36398	−8.029	−8.029	−38.61
HSPA8–spike protein	NSC281245	−5.285	−5.285	−36.65
HSPA8–spike protein	NSC11926	−4.120	−4.120	−27.16
HSPA8–spike protein	NSC44175	−2.796	−2.798	1.61

## Data Availability

The data utilized to substantiate the study’s conclusions are providedin this article.

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
