# Peer review of "In Silico Discovery and Evaluation of Inhibitors of the SARS-CoV-2 Spike Protein–HSPA8 Complex Towards Developing COVID-19 Therapeutic Drugs"

_viruses, 2024, doi:10.3390/v16111726_

Round 1

Reviewer 1 Report

Comments and Suggestions for Authors

The paper of Navhaya et al. Is devoted to evaluation of formation of complexes between the spike and HSPA8 proteins and the ligands and above indicated proteins. The theme is not new but very popular due to practical role of spike protein in SARS-COVID disease. The results obtained by the authors are rather interesting and may be useful in a treatment of the next level medicine against SARS-COVID. The results are well verified (two different methods are used for the evaluations) and support the conclusions. I expect the manuscript can be published with minor revisions

(see below)

1) Presentations of figures are not clear. Please, clarify

1a) whats means the green color in Fig. 1?

1b) what means the yellow small circle with symbol B in Fig. 3?

1c) what mean colors different fro gray in Figs. 4 and 8?

2) The authors claimed “NSC44175 showed the lowest docking score of –2.873 kcal/mol, Glide Gscore of –2.873 kcal/mol and the lowest binding free energy” (page 14). However for negative values as lower is the absolute value of the free energy as HIGHER the energy.

3) Results obtained by the scoring methods and the MM-GBSA approach are slightly different.

It would be helpful, if the authors will provide few comments indicating why they prefer the results of the scoring methods for the final conclusions.

Author Response

Reviewer 1’s comments

We are grateful for your thoughtful review of our manuscript, Navhaya et al. We appreciate the recognition of the significance of our work in evaluating the complexes formed between the spike and HSPA8 proteins and the ligands. We're glad to hear that you found our results interesting and valuable for future SARS-CoV-2 treatment strategies. Based on your comments, the following comments and revisions were made on our end.

  1. Comment 1
    1. The green portions of the structures are from the default colouring of PyMOL software that was used in visualizing the modelled 3D structures. Generally, when visualizing a 3D protein structure, PyMOL colours the structure green. In our efforts to differentiate the essential structures of each protein, we decided to colour the other regions in different colours from the default one. In a nutshell, the green-coloured portion of each protein indicates the rest of the protein that is not known for its specific role or function or the portions that were not of interest in our article.
    2. The β (beta) symbol in Fig. 3 is often used in structural biology to denote the beta-carbon (βC) of amino acids in a protein structure involved in the interaction of proteins. We have included the meaning of such representation as shown in lines 330
    3. In Fig 4 and 8, the grey colouring is the default colouring of the BioLuminate/Maestro Schrodinger software package when visualising a 3D structure. In Fig 4, the red, blue and yellow sections represent the identified binding site in each individual. However, the binding site has been identified correctly in line 335 for clarity's sake. In Fig 8, two different proteins, HSPA8 and spike protein, that make up the protein complex were coloured differently for easy identification. However, they were labelled next to each protein.
    4. Comment 2: This statement's error was rectified accordingly (lines 424-425).
    5. Comment 3: In formulating a conclusion, we did not consider only the docking scores. We also considered MM-GBSA values, which further supported the results obtained from the docking simulations. We also targeted small molecules that target the spike protein individually and in complex with human HSPA8. With this mentioned, the conclusion was revised (lines 640-666).

Reviewer 2 Report

Comments and Suggestions for Authors

Dear Authors,

Thank you for submitting your article titled "In-silico discovery and evaluation of inhibitors for HSPA8:SARS-CoV-2 spike protein complex towards developing COVID-19 therapeutic drugs." Your work delves into the in silico assessment of the binding ability of the SARS-CoV spike protein and HSPA8, as well as the potential of several small molecules to disrupt this binding to propose them as potential antiviral therapies.

This manuscript builds on previously published work on in silico analysis of the potential role of Heat Shock Proteins in the replication and development of SARS-CoV virus, particularly the interaction of non-structural proteins ([5] -Yamkela M, Sitobo and Makhoba, 2023) and structural spike proteins ([7] - Navhaya, et al, 2024). The current work is based on the hypothesis of the membrane location of HSPA8 and its potential involvement in the induction of the endocytosis and entry steps of the viral life cycle. Authors  used in silico modeling to assess HSP8 and SARS-CoV-2 spike proteins, as well as the binding capacity of several small molecules to these proteins individually and in complex. However, the focus on docking scores and binding affinities without strong experimental observations raises my main  concerns. While in silico modeling is valuable, it should be complemented by experimental studies or used to support experimental results. The connection between observations and real experimental results should be backed by strong supportive references. Additionally, the presentation style tends to use statement-like phrases with not very evident or indirect references, often citing the authors' own two previous publications. The extensive use of references, especially to their own work, without clear relation to the statement in the phrase complicates the review process.

The absence of line numbering in the manuscript made it more challenging to demonstrate discrepancies in the references used.  I would strongly recommend to review current references and to refer to more appropriate sources for example:

-       the reference you provide on page 2 (top line) to the structure of SARS-CoV [5] by Yamkela M et al, 2023, would be more appropriate if it referenced a review or major source of information about the virus's structure, rather than your previous publication. Similarly, the reference to [6], which discusses entry inhibitors and offers excellent references regarding virus structure, would benefit from being linked to more appropriate sources.

-       On the same page, the information discussed in not the information from Reference [7]. It was not explained why among all HSP proteins tested in manuscript [5] specifically was chosen HSPA8?

-       I was not able to find information about specifically those 4 compounds in reference [11], please provide similar names or better ID to align them with the source.

Additional specific recommendations for improving the manuscript include reformatting of the  figure 1 the legend in the colored boxes is very hard to read. Figure 2 the Ramachandran plot statistics table are very hard to read it is making sense to put them separate in a formatted table.

Please review the comments provided in the PDF of the manuscript. It is important to address these concerns, refine the presentation style, and ensure accurate and relevant referencing to enhance the scientific  quality of the manuscript.

Sincerely

Author Response

Reviewer 2’s comments

Thank you for your thorough evaluation of our manuscript. Your recognition of our work is greatly appreciated. Please be aware that this research project is still ongoing. We decided to publish the results of our in-silico findings. The research project is divided into in-silico and in-vitro work, in which the in-vitro work validates our findings in the in-silico phase of our work. We are busy acquiring the required consumables to start with the in-vitro work.  All comments mentioned in the PDF of the manuscript were attended to accordingly, accompanied by a brief description of what or how the issue mentioned in the comment was resolved.

Reviewer 3 Report

Comments and Suggestions for Authors

The article is clear in presenting its objectives, methods, and in explaining the results obtained from the conducted docking studies. However, in the introduction section, I believe it lacks a brief focus on why, among many possible molecules, the ones discussed in the article were chosen. In section 2.2, "Identification and selection of potential small molecules," it is indeed mentioned that these are already known and validated molecules and that all four of them have known inhibitory properties against the SARS-CoV-2 Mpro protein. However, this does not justify the decision to conduct docking studies of these molecules with the target proteins described in the article.

Author Response

Reviewer 3’s comments

Thank you for your valuable feedback. We appreciate your suggestions regarding the introduction. To address this, we have revised the introduction and included a brief explanation of the rationale behind the selection of the small molecules (lines 128-137; lines 149 -161) and the justification for the selection of the protein HSPA8 (lines 85-115)

Round 2

Reviewer 2 Report

Comments and Suggestions for Authors

Dear Authors

Thank you for your attention to the corrections and incorporating the recommendations made in the review. The revisions have significantly enhanced the scientific quality of the manuscript and have brought it in line with higher scientific standards. There are still some sections with rough phrases that may require further review from an English language perspective. For example:

Line 62-63: “From the reviewed 62 literature, the pathogenesis of hCoVs’ understanding is limited, and the …” – the pathogeneses of understating ?

Line 65 -67: “The pro-infection and antiviral activities present opportunities  for developing antiviral treatments and therapies via exploiting their immune activities or inhibiting the molecular chaperones that have pro-infection activities” – It is very hard for understanding.

Line 85: “Literature dictates that HSPA8 have been implicated in the life…- literature is not dictating anything.

Line 189-190:  “For drug preparation, the small molecules were obtained directly from PubChem in ‘.SDF’ format and converted to ‘.mol2’ format using PyMOL software.” – the molecules were not obtained in .sdf format but their structure was .

Comments on the Quality of English Language

few examples of the unclear or inaccurate phrases: 

Line 62-63: “From the reviewed 62 literature, the pathogenesis of hCoVs’ understanding is limited, and the …” – the pathogeneses of understating ?

Line 65 -67: “The pro-infection and antiviral activities present opportunities  for developing antiviral treatments and therapies via exploiting their immune activities or inhibiting the molecular chaperones that have pro-infection activities” – It is very hard for understanding.

Line 85: “Literature dictates that HSPA8 have been implicated in the life…- literature is not dictating anything.

Line 189-190:  “For drug preparation, the small molecules were obtained directly from PubChem in ‘.SDF’ format and converted to ‘.mol2’ format using PyMOL software.” – the molecules were not obtained in .sdf format but their structure was .

Author Response

Thank you very much for your help, all comments have been attended to.